# Learning to Adapt to Evolving Domains

**Hong Liu**[†], **Mingsheng Long**[‡*], **Jianmin Wang**[‡], **Yu Wang**[†]

[‡]School of Software, KLiss, BNRist, Tsinghua University
[†]Department of Electronic Engineering, Tsinghua University
h-l17@mails.tsinghua.edu.cn
{mingsheng,jimwang,yu-wang}@tsinghua.edu.cn

## Abstract

Domain adaptation aims at knowledge transfer from a labeled source domain to an unlabeled target domain. Current domain adaptation methods have made substantial advances in adapting discrete domains. However, this can be unrealistic in real-world applications, where target data usually come in an online and continually evolving manner, posing challenges to classic domain adaptation paradigm: (1) Mainstream domain adaptation methods are tailored to stationary target domains, and can fail in non-stationary environments. (2) Since the target data arrive online, the model should also maintain competence on previous target data, i.e. adapt without forgetting. To tackle these challenges, we propose a meta-adaptation framework which enables the learner to adapt to continually evolving target domains without forgetting. Our framework consists of two components: a meta-objective of learning representations to adapt to evolving domains, enabling meta-learning for unsupervised domain adaptation; and a meta-adapter for learning to adapt without forgetting, reserving knowledge from previous target data. Experiments validate the effectiveness our method on evolving domain adaptation benchmarks.

## 1   Introduction

Many machine learning applications require consistent performance across datasets of different underlying data distributions. A plausible solution to this problem is domain adaptation, which bridges the dataset shift and circumvents manual annotation for new tasks [26, 25]. By learning features transferring well from a labeled source domain to an unlabeled target domain, domain adaptation facilitates knowledge transfer and mitigates the harmful effect of shift in data distributions.

In this paper, we consider a more realistic setting of domain adaptation — *adapting to evolving target domains*. Imagine a self-driving agent with a scene recognition system trained on scenes from a stationary condition (the source domain). When deployed in real world, the environment can vary in a continually evolving way, such as from day to night, and from shine to rain. Therefore, we hope the agent gradually adapts to the environment shift and performs consistently well on scenes from all environments (the evolving target domain). Another restriction we are confronted with is the limited computational resources of the agent when deployed. Thus, we may learn representations in the factory, and deploy only light-weight model to the cars when adapting to new target data in the real world. Following the terms of meta-learning [4], we formulate this setting as evolving domain adaptation (**EDA**): (1) we have access to adequate labeled examples from the source domain, and part of the target unlabeled data from a target domain *evolving over time* in the meta-training phase, (2) new target data of the meta-testing phase arrive sequentially online from the same evolving target distribution and cannot be stored, and (3) after adapting to these new target data sequentially, we test the model on all target data of meta-testing. An example of the EDA problem is provided in Figure 1.

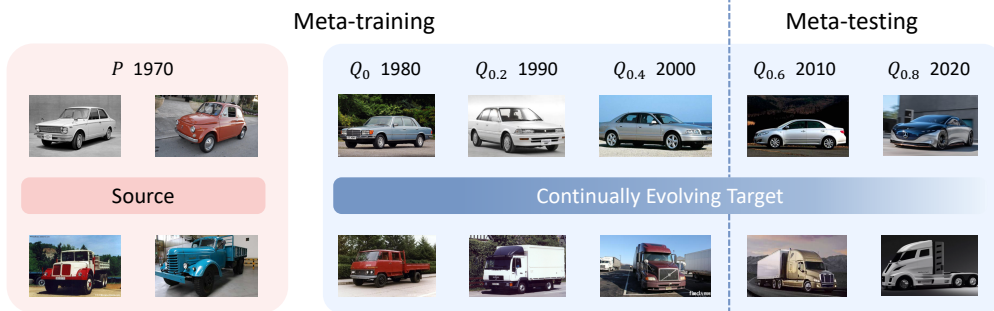

Figure 1: **Problem Setup of Evolving Domain Adaptation (EDA)**: The source domain is labeled and stationary, while the target domain is unlabeled and evolves with time. In the meta-training phase we have access to part of the target data. In meta-testing, new target data come sequentially. We hope the model performs consistently well on all target data in meta-testing after adaptation.

The evolving target domain poses obstruction to knowledge transfer. Mainstream domain adaptation methods are tailored to adapting discrete source and target domains. [6, 19, 35, 31, 20, 40] learn domain-invariant representations via domain adversarial training. This line of works rely on a domain discriminator to distinguish two discrete domains, and cannot apply to evolving target domain directly.

Moreover, since the model adapts to new target data sequentially without storing the data, EDA faces another challenge, catastrophic forgetting [23]. Current neural networks are prone to overwriting older knowledge while adapting to new tasks [14]. In EDA, we hope the model can retain competence on previous target data when adapting to new target data. Admittedly, a line of works have been proposed to ameliorate catastrophic forgetting in supervised learning. Regularization methods [14, 17, 39] prevent features with important knowledge from deviating too much on the new tasks, but calculating "importance" relies on supervision from tasks. Replay methods [28, 33, 38, 30] need storing samples from previous tasks or using large generative models [10], which violate the setting of EDA and are unrealistic with the limited resources of online devices such as aforementioned self-driving agents.

Aiming to tackle both challenges, this paper proposes Evolution Adaptive Meta-Learning (**EAML**), a meta-adaptation framework to adapt to continually evolving target domain without forgetting. Recent advances in meta-learning [4, 5, 27] have been gaining momentum. Through an explicit learning-to-learn paradigm, the meta-learner is able to learn transferable features for downstream tasks [13, 29], which is critical in few-shot learning and transfer learning. Similarly, we can specify a meta-training procedure of learning to adapt without forgetting explicitly. Concretely, our model comprises two components: (1) a meta-objective to learn representations for adapting to continually evolving target data, and (2) a meta-adapter for adapting to current target without forgetting the previous target. The EAML framework is naturally suitable for EDA setting: In the meta-training phase we obtain meta-representations and meta-adapters specified for evolving domain shift. When adapting to new target data in the meta-testing phase, the meta-representations and meta-adapters are fixed, with only light-weight adapters trained on small batches of data. We validate the proposed approach on evolving domain adaptation benchmarks. Results demonstrate the effectiveness of the proposed approach of addressing both evolving domain shift and catastrophic forgetting.

## 2  Evolving Domain Adaptation

Suppose we are provided with a source distribution $P(x, y)$ and an evolving target distribution $Q_t(x, y)$, $t \in [0, 1]$. For instance, if the target underlying distribution evolves with years, $t$ will be associated with a particular year. To quantify the continually evolving nature of $Q_t(x, y)$, typically $d(Q_{t_1}, Q_{t_2}) > 0$ for some distribution distance $d$. We further assume $\lim_{\Delta t \to 0} d(Q_t, Q_{t+\Delta t}) = 0$ as the continuity of the evolvement. See Figure 1 for this example. The goal of EDA is to learn a model $f_\theta$ parametrized by $\theta$ with consistently good performance over the continually evolving target $Q_t$, i.e.

$$\min_\theta \mathbb{E}_{t \sim U(0,1)} \mathbb{E}_{(x,y) \sim Q_t} L(f_\theta(x), y) = \min_\theta \int_0^1 \mathbb{E}_{(x,y) \sim Q_t} L(f_\theta(x), y) \mathrm{d}t. \qquad (1)$$

$L$ is a loss function. For simplicity, we consider binary classification and 0-1 loss in this section. We have access to $n_s$ source labeled data points $S = \{\mathbf{X}_s, \mathbf{Y}_s\} = \{x_{s,i}, y_{s,i}\}_{i=1}^{n_s}$ sampled i.i.d. from the

source distribution $P$. Target unlabeled data points come in small batches sequentially,

$$T = \{\mathbf{X}_{t_1}, \mathbf{X}_{t_2} \cdots \mathbf{X}_{t_n}\},$$

forming a trajectory of evolving target. Here $t_1 < t_2 < \cdots < t_n$ is an ordered sequence of $n$ i.i.d. samples from uniform distribution $U(0,1)$, and each $\mathbf{X}_{t_i}$ has $n_{t_i}$ i.i.d. samples from $Q_{t_i}$. Note that this problem fundamentally differs from classic domain adaptation. The underlying difference causes significant degradation in performance if applying standard domain adaptation techniques to EDA.

To solve the EDA problem with target trajectories, we need the target domain to evolve steadily to facilitate knowledge transfer. Recall the mathematical tools in analyzing generalization bound of classic domain adaptation [1, 22]. Denote by $d_{\mathcal{H}\Delta\mathcal{H}}$ the $\mathcal{H}\Delta\mathcal{H}$-divergence, $d_{\mathcal{H}\Delta\mathcal{H}}(P, Q_t) = \sup_{\theta,\theta'} |\mathbb{E}_P L(f_\theta(x), f_{\theta'}(x)) - \mathbb{E}_{Q_t} L(f_\theta(x), f_{\theta'}(x))|$, which quantifies the discrepancy between the source and target distributions. The adaptability measuring the possibility of cross-domain learning is $\lambda_t = \min_\theta [\mathbb{E}_P L(f_\theta(x), y) + \mathbb{E}_{Q_t} L(f_\theta(x), y)]$. We extend this analysis to the EDA problem.

**Theorem 1.** *We further assume $d_{\mathcal{H}\Delta\mathcal{H}}(Q_{t_1}, Q_{t_2}) \leq \alpha |t_1 - t_2|$ holds with constant $\alpha$ for $t_1, t_2 \geq 0$. Then for any $\theta$, with probability at least $1 - \delta$ over the sampling of target trajectory $t_1, t_2 \cdots t_n$,*

$$\mathbb{E}_t \mathbb{E}_{Q_t} L(f_\theta(x), y) \leq \mathbb{E}_P L(f_\theta(x), y) + \frac{1}{n} \sum_{i=1}^{n} [d_{\mathcal{H}\Delta\mathcal{H}}(P, Q_{t_i})] + \mathbb{E}_t \lambda_t + O\left(\frac{\alpha}{\delta n}\right).$$

**The role of $\alpha$ in EDA.** Intuitively, $\alpha$ indicates the rate of evolvement of the target domain $Q_t$. A reasonably small $\alpha$ means $Q_t$ is evolving evenly, and neighboring target data share knowledge. $n$ is the length of target trajectory $T$. With small $\alpha$ and large $n$, i.e. if the target domain is evolving steadily and the target trajectory is of sufficient length, we can solve the EDA problem by adapting source and target trajectories. To this end, EDA methods should consider adapting to random samples of target trajectories instead of adapting to the single target in order to generalize on evolving target data. Furthermore, EDA methods should also learn representations to capture and harness the evolvement of target domain, i.e. make $\alpha$ sufficiently small on learned representations.

Another critical issue of EDA is that the target data come online and cannot be stored in meta-testing. Since neural networks are prone to overwriting previous knowledge in new tasks [23], adapting to current target data inevitably results in forgetting knowledge on previous target. To address this problem, we should design specific networks to mitigate catastrophic forgetting in EDA.

Motivated by the above insights, we propose Evolution Adaptive Meta-Learning (EAML), comprising two components tailored to continually evolving domain shift and catastrophic forgetting respectively.

## 3 Evolution Adaptive Meta-Learning

Applying discrete domain adaptation methods directly to evolving target domain cannot solve EDA properly. First, discrete DA neglects the evolvement of the target domain, which exerts significant impact on EDA as pointed out by Theorem 1. Adapting to an intermediate target domain before distant target domain performs better than adapting to them as a whole target [9]. Thus, we should specify a training strategy to harness the evolvement of target data. Another factor lies in that the target data come online sequentially, though we hope the model to perform well on all target data. To retain competence on previous target data, we should overcome catastrophic forgetting.

Recent advances in meta-learning [4, 27] have enabled learning transferable representations adapting to following tasks rapidly. Through an explicit learning-to-learn paradigm, the model can learn representations specified for downstream tasks such as few-shot learning. Inspired by this, if we learn to adapt explicitly, we can obtain a meta-representation specified for EDA. Similarly, we can also learn a meta-adapter to tackle catastrophic forgetting through learning not to forget. See Figure 2(a) for an overview of the proposed EAML framework.

### 3.1 Learning a Meta-Representation for Continually Evolving Target

Recall that Model-Agnostic Meta-Learning (MAML) [4] learns transferable features by minimizing error on the support set in the inner loop and minimizing error on the query set in the outer loop. We can learn a representation for adapting to evolving target similarly. Denote by $h_\theta$ the representation function parametrized by $\theta$, $g_\phi$ the adapter parametrized by $\phi$, and $c_W$ the task classifier with parameters $W$. Then the model can be expressed as the composite function $f_{\theta,\phi,W} = c_W \circ g_\phi \circ h_\theta$. Taking the meta-training protocol [4], we sample source data $S = \{\mathbf{X}_s, \mathbf{Y}_s\} = \{x_{s,i}, y_{s,i}\}_{i=1}^{n_s}$, a

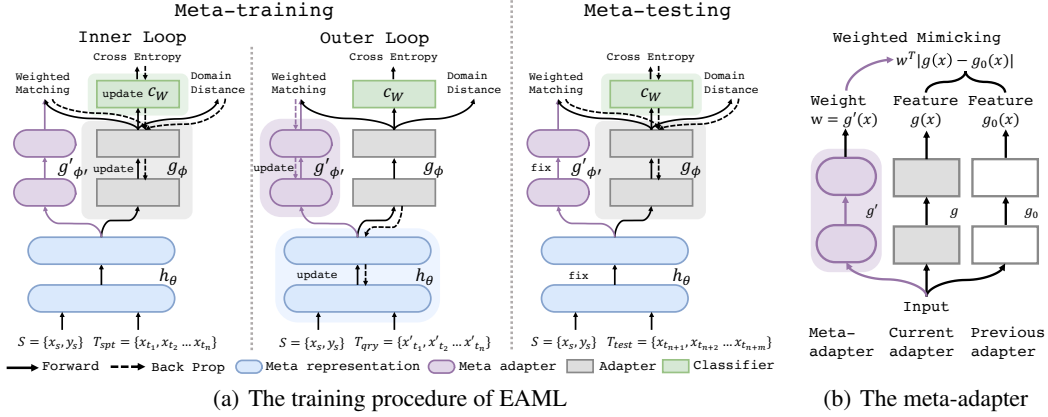

Figure 2: **(a) Training procedure of EAML.** In the inner loop, the adapter and the task classifier are updated. In the outer loop, the meta-representation improves evolving adaptation, while the meta-adapter overcomes forgetting. In meta-testing, both the meta-representation and the meta-adapter are fixed, with only the light-weight adapter and task classifier trained on new target data. **(b) Training with the meta-adapter.** We mimic the features of adapters across time. The meta-adapter learns weights for feature matching in different dimensions to retain useful features and overcome forgetting.

target trajectory from the support set as $T_{\text{spt}} = \{\mathbf{X}_{t_1}, \mathbf{X}_{t_2} \cdots \mathbf{X}_{t_n}\}$, and a target trajectory from the query set as $T_{\text{qry}} = \{\mathbf{X}'_{t_1}, \mathbf{X}'_{t_2} \cdots \mathbf{X}'_{t_n}\}$. Note that each $t_i$ in the support and query set is identical.

**Train adapter and classifier in the inner loop.** In the inner loop, we train the adapter $g_\phi$ and task classifier $c_W$ to adapt to evolving target sequentially on the current representation $f_\theta$. Thus, the representation $f_\theta$ is fixed in the inner loop. After initializing $g$ and $c$ with $\phi_0$ and $W_0$, we adapt them to the target samples in $T_{\text{spt}}$ sequentially with SGD: for $i = 0, 1, \cdots, n-1$,

$$(\phi, W)_{i+1} \leftarrow (\phi, W)_i - \eta_{\text{in}} \nabla_{(\phi, W)} \left[ L(f_{\theta, \phi_i, W_i}(\mathbf{X}_s), \mathbf{Y}_s) + d(f_{\theta, \phi_i, W_i}(\mathbf{X}_s), f_{\theta, \phi_i, W_i}(\mathbf{X}_{t_i})) \right], \tag{2}$$

where $\eta_{\text{in}}$ is the learning rate of the inner loop. In classification setting, $L$ is implemented as cross-entropy loss. We use joint maximum mean discrepancy (joint MMD) [21] as domain discrepancy $d$, though other discrepancy measures can also fit in our framework.

**Train the meta-representation in the outer loop.** In the outer loop, we require the representation to make the adaptation in the inner loop more effective. A direct approach is to update the representation $f_\theta$ to minimize the EDA loss on the query set following Equation (1). However, we have no access to target labels. Inspired by Theorem 1, we replace the original EDA loss with its upper bound, and update $\theta$ to control the upper bound. Note that we have shown the critical role of the evolving rate $\alpha$ in EDA problem. In order to learn a representation tailored to EDA, we should take $\alpha$ into consideration. We use $\max_i d(f_{\theta, \phi_n, W_n}(\mathbf{X}'_{t_{i-1}}), f_{\theta, \phi_n, W_n}(\mathbf{X}'_{t_i}))$ as an approximation of $\alpha$, and update $\theta$ by

$$\theta \leftarrow \theta - \eta_{\text{out}} \nabla_\theta L[(f_{\theta, \phi_n, W_n}(\mathbf{X}_s), \mathbf{Y}_s) + \frac{1}{n} \sum_{i=1}^{n} d(f_{\theta, \phi_n, W_n}(\mathbf{X}_s), f_{\theta, \phi_n, W_n}(\mathbf{X}'_{t_i}))$$
$$+ \max_i d(f_{\theta, \phi_n, W_n}(\mathbf{X}'_{t_{i-1}}), f_{\theta, \phi_n, W_n}(\mathbf{X}'_{t_i}))], \tag{3}$$

where $\eta_{\text{out}}$ denotes the learning rate of the outer loop.

### 3.2 Learning a Meta-Adapter to Overcome Forgetting

We have obtained a meta-representation for adapting to continually evolving target. Nonetheless, when adapting to new trajectory of target $T_{\text{test}}$ in the meta-testing, the representation $f_\theta$ is fixed, with only $g_\phi$ and $c_W$ adapting to target data online. To this end, we should also design the adapter $g_\phi$ to avoid overwriting knowledge learned by the adapter on the previous target. This can be achieved by introducing a meta-adapter $g'_{\phi'}$ to the original adapter $g_\phi$.

**Mimicking intermediate features with meta-adapter in the inner loop.** If an adapter is well-trained on a previous target, its intermediate features contain useful knowledge for that target data. Thus, matching the intermediate features when adapting to new target helps overcome forgetting of

**Algorithm 1** Meta-Training of Evolution Adaptive Meta-Learning (EAML)

1: **Input:** the source domain $P$ and the evolving target domain $Q_t$.
2: **Output:** learned representations $f_\theta$ and meta-adapter $g'_{\phi'}$.
3: **for** $t = 0$ **to** MaxIter **do**
4:    Initialize adapter $g_\phi$ and task classifier $c_W$; Sample source data $S = \{\mathbf{X}_s, \mathbf{Y}_s\}$ from $P$.
5:    Sample target trajectories $T_{\text{spt}} = \{\mathbf{X}_{t_1}, \mathbf{X}_{t_2} \cdots \mathbf{X}_{t_n}\}$ and $T_{\text{qry}} = \{\mathbf{X}'_{t_1}, \mathbf{X}'_{t_2} \cdots \mathbf{X}'_{t_n}\}$ from $Q_t$.
6:    **for** $i = 0$ **to** $n$ **do**
7:       In the inner loop, update the classifier $c_W$ and the adapter $g_\phi$:

$$(\phi, W)_{i+1} = (\phi, W)_i - \eta_{\text{in}} \nabla_{(\phi, W)} [L(f_{\theta, \phi_i, W_i}(\mathbf{X}_s), \mathbf{Y}_s)$$
$$+ d(f_{\theta, \phi_i, W_i}(\mathbf{X}_s), f_{\theta, \phi_i, W_i}(\mathbf{X}_{t_i})) + \lambda L_m(\phi', \phi_i, \phi_{i-1})].$$

8:    **end for**
9:    In the outer loop, update the meta-representation $f_\theta$ and the meta-adapter $g'_{\phi'}$:

$$(\theta, \phi') \leftarrow (\theta, \phi') - \eta_{\text{out}} \nabla_{\theta, \phi'} L[(f_{\theta, \phi_n, W_n}(\mathbf{X}_s), \mathbf{Y}_s) + \frac{1}{n} \sum_{i=1}^n d(f_{\theta, \phi_n, W_n}(\mathbf{X}_s), f_{\theta, \phi_n, W_n}(\mathbf{X}'_{t_i}))$$
$$+ \max_i d(f_{\theta, \phi_n, W_n}(\mathbf{X}'_{t_{i-1}}), f_{\theta, \phi_n, W_n}(\mathbf{X}'_{t_i}))].$$

10: **end for**

---

the old ones. Yet not all the intermediate features are critical to the previous target. A more plausible solution is learning to assign different weights for different features.

Suppose the $l$-th intermediate layer of the adapter is $g_{l,\phi_l} : \mathcal{R}^d \to \mathcal{R}^m$ parametrized by $\phi_l$, $g_{l,\phi_l^{\text{prev}}}$ is $l$-th layer of the previous target, and the features in a mini-batch are $x \in \mathcal{R}^{n \times d}$. Then the weighted feature matching loss is $\sum_{j=1}^n w_j^\top \|g_{l,\phi_l}(x_j) - g_{l,\phi_l^{\text{prev}}}(x_j)\|$, where $w_j \in \mathcal{R}^m$ is the weight of sample $x_j$. To facilitate knowledge retaining, we introduce the meta-adapter $g'_{l,\phi'_l}$, $w_j = g'_{l,\phi'_l}(x_j)$. Thus, the total loss is the sum of the weighted feature matching loss in each layer,

$$L_m(\phi', \phi, \phi^{\text{prev}}) = \sum_l \sum_j g'_{l,\phi'_l}(x_j)^\top \|g_{l,\phi_l}(x_j) - g_{l,\phi_l^{\text{prev}}}(x_j)\|.$$

We describe the weighted feature matching with the meta-adapter in Figure 2(b). In the inner loop, we learn the adapter $g_\phi$ and the classifier $c_W$ with the weight generated by $g'_{\phi'}$. Concretely, we add the feature matching loss $L_m$ to the original loss in Equation (2), and fix the meta-adapter $g'_{\phi'}$.

**Learning the meta-adapter in the outer loop.** In the outer loop, we need to update the meta-adapter to make the weighted feature mimicking preserve more knowledge on previous target data. Similar to the strategy of learning the meta-representations, we use the same loss function as Equation (3), but we calculate the gradient w.r.t. both $f_\theta$ and $g'_{\phi'}$.

We sum up the complete training procedure of the proposed method in Algorithm 1 and Figure 2(a). In the inner loop, the adapter $g_\phi$ and task classifier $c_W$ are trained to adapt to target data sequentially. In the outer loop, both meta-representation $f_\theta$ and meta-adapter $g'_{\phi'}$ are updated to facilitate adaptation to evolving target without forgetting. During meta-training, we learn both meta-representation and meta-adapter offline. When exposed to new target trajectory in meta-testing, $f_\theta$ and $g'_{\phi'}$ are fixed and only $g_\phi$ and $c_W$ are updated, resulting in a light-weight model when deployed to scenarios short of computation resources such as self-driving agents.

## 4 Experiments

In this section, we evaluate our method with evolving domain datasets in different scenarios. Details on datasets and implementation are deferred to appendix[2].

### 4.1 Datasets

**Rotated MNIST:** This dataset consists of MNIST digits of various rotations. This is similar to the protocol of [3], but they only consider discrete rotations and each domain has 60000 training samples.

Table 1: Classification Accuracy (%) on rotated MNIST dataset.

| Method | 120° | 126° | 132° | 138° | 144° | 150° | 156° | 162° | 168° | 174° | Avg. |
|---|---|---|---|---|---|---|---|---|---|---|---|
| Source Only | 17.60 | 19.29 | 22.50 | 24.14 | 26.49 | 29.48 | 31.06 | 32.26 | 33.73 | 33.25 | 26.98 |
| DANN [6] | 18.92 | 21.65 | 24.32 | 27.63 | 29.76 | 32.01 | 33.92 | 36.23 | 36.68 | 36.93 | 29.81 |
| JAN [21] | 20.37 | 22.53 | 25.15 | 27.53 | 29.89 | 31.40 | 32.99 | 33.85 | 35.77 | 37.13 | 29.66 |
| JAN Merge | 20.20 | 21.71 | 25.75 | 29.16 | 33.27 | 37.19 | 40.04 | 40.39 | 39.67 | 38.71 | 32.60 |
| CDAN Merge [20] | 20.50 | 23.64 | 25.73 | 29.65 | 31.44 | 37.05 | **40.17** | 40.02 | 41.16 | 39.28 | 32.87 |
| MAML [4] | 22.75 | 25.11 | 28.90 | 30.40 | 32.62 | 34.56 | 35.14 | 36.55 | 37.31 | 38.30 | 32.16 |
| CMA [11] | 21.82 | 23.65 | 26.48 | 29.48 | 32.05 | 34.99 | 35.08 | 36.34 | 38.33 | 39.25 | 31.75 |
| DANN+EWC [14] | 20.39 | 24.19 | 28.50 | 30.10 | 32.48 | 35.75 | 36.23 | 38.47 | 38.63 | 38.05 | 32.27 |
| EAML (rep) | 20.91 | 23.91 | 26.23 | 29.74 | 33.37 | 36.15 | 38.93 | **41.09** | 42.05 | **43.30** | 33.57 |
| EAML (ada) | 22.34 | 26.07 | **30.83** | 31.20 | 32.25 | 36.36 | 37.69 | 38.75 | 38.31 | 39.13 | 33.19 |
| EAML (full) | **24.69** | **27.48** | 30.16 | **32.79** | **34.88** | **37.35** | 39.25 | 40.96 | **42.45** | 42.27 | **35.23** |
| Replay Oracle | 24.35 | 26.21 | 30.33 | 31.89 | 33.02 | 35.87 | 37.98 | 39.66 | 41.40 | 42.21 | 34.28 |

In our modified protocol, the rotation of target domain is continuous $0 - 180°$. Images with rotation $0°$ belong to the labeled source domain. For each rotation, we have access to 100 samples of images. In meta-training, we use rotation $0 - 60°$. We randomly sample a trajectory $T = \{\mathbf{X}_{t_1}, \mathbf{X}_{t_2} \cdots \mathbf{X}_{t_{10}}\}$, $t_i \in [0, 60]°$, formulating an EDA problem. In meta-testing, we test the model's performance online with trajectory $T = \{\mathbf{X}_{120°}, \mathbf{X}_{126°} \cdots \mathbf{X}_{174°}\}$. Note that in meta-testing, we have only 100 training samples for each rotation, making this task very challenging.

**Evolving Vehicles:** This dataset contains sedans and trucks from the 1970s to 2010s (See Figure 1), which involves more complex continuous domain shift compared to rotated MNIST. For each decade, we collect 100 sedans and 100 trucks. Images from 1970 to 1975 are used as the labeled source domain. The continually evolving target domain contains vehicles from 1980 to 2020. In meta-training, We randomly sample a trajectory $T = \{\mathbf{X}_{t_1}, \mathbf{X}_{t_2} \cdots \mathbf{X}_{t_6}\}$ from 1980 to 1995, formulating an EDA problem. In meta-testing, we adapt to trajectory $T = \{\mathbf{X}_{2000}, \mathbf{X}_{2005} \cdots \mathbf{X}_{2015}\}$ sequentially online and test on all images in $2000 \sim 2020$.

**Caltran:** This is a real-world dataset of images captured by a camera at an intersection over time. It contains images captured at an interval of 3 minutes over two weeks, formulating a challenging continually evolving target, since it includes changes in time, illumination, weather, etc. [11] evaluated the performance on this dataset, yet they tested the model online, without accounting for catastrophic forgetting. We modify their original setup as follows: Images from the first 5 hours (100 in total) are used as the labeled source domain. The continually evolving target domain contains images in the following two weeks. In meta-training, We sample a trajectory $T = \{\mathbf{X}_{t_1}, \mathbf{X}_{t_2} \cdots \mathbf{X}_{t_{100}}\}$ in the first week. In meta-testing, we adapt a trajectory of length 100 drawn randomly from the images in the second week and test the performance on all images from the second week.

### 4.2 Implementation

We implement our method on PyTorch. We adopt cross-entropy loss for classification tasks. For the domain discrepancy, we use joint MMD [21]. Suppose the feature representations and the logits of the source domain and the target domain are $\{x_i^p, \hat{y}_i^p\}_{i=1}^{n_p}$ and $\{x_i^q, \hat{y}_i^q\}_{i=1}^{n_q}$, respectively. We are provided with the kernel function $k_x(\cdot, \cdot)$ and $k_y(\cdot, \cdot)$ for the feature and output spaces. Denote by $k((x_1, y_1), (x_2, y_2)) = k_x(x_1, x_2) \cdot k_y(y_1, y_2)$ the joint kernel function, then the joint MMD is computed as the squared distance between empirical kernel means,

$$d(P, Q) = \mathbb{E}k((x^p, \hat{y}^p), (x^p, \hat{y}^p)) + \mathbb{E}k((x^q, \hat{y}^q), (x^q, \hat{y}^q)) - 2\mathbb{E}k((x^p, \hat{y}^p), (x^q, \hat{y}^q)).$$

We use $\ell_2$ loss for feature matching. The importance of the feature matching loss in the total loss is set by cross-validation. We use SGD with 0.9 momentum and $5 \times 10^{-4}$ weight decay. The learning rates of the inner loop and the outer loop are set to 0.01 and 0.001 respectively. For rotated MNIST, we use LeNet [16] as the backbone. The meta-representation $f_\theta$ includes two convolutional layers. The adapter is a two-layer fully-connected network with ReLU activations. To avoid the impact of image pre-processing, we only normalize the input to zero-mean and uni-variance without random rotation or flip. For Evolving Vehicles and Caltran, we use a six-layer convolutional network as the backbone. The meta-representation $f_\theta$ includes four convolutional layers, the same as the conv-4 network in [4]. The adapter is a two-layer convolutional network with ReLU activations. We carry out each experiment 3 times and report the mean accuracy and standard deviation.

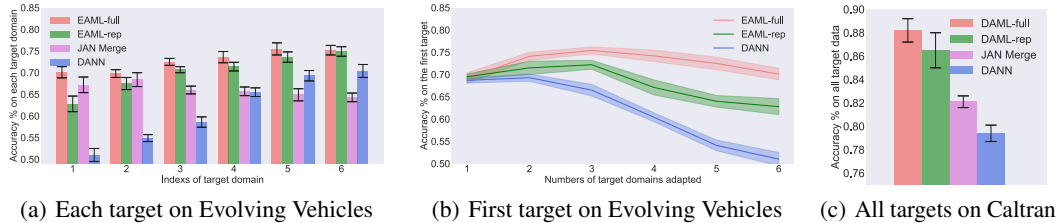

(a) Each target on Evolving Vehicles     (b) First target on Evolving Vehicles     (c) All targets on Caltran

Figure 3: **Results on Evolving Vehicles and Caltran.** (a) Accuracy of each target domain after adapting to them sequentially online on Evolving Vehicles. (b) Accuracy of the first target domain after adapting to following target data on Evolving Vehicles. (c) Accuracy of all target data in meta-testing on Caltran.

## 4.3 Baselines

**Source Only.** Simply train on the source domain and test on the target without adaptation.

**DANN [6] and JAN [21].** These two represent classic domain adaptation methods for discrete target domains. We apply them to the EDA problem as follows. We first train the model with labeled source dataset and all the target data available offline in meta-training. Then we adapt both methods to the evolving target data sequentially in meta-testing and test their performance.

**JAN Merge and CDAN Merge.** CDAN [20] is a strong classic domain adaptation baseline. We merge the continually evolving target available in both meta-training and meta-testing as a single target and train JAN or CDAN on it. We test its performance on the target data of meta-testing. Note that merging the target data and training the model on it offline actually violates the setting of EDA, and both methods can take benefit from such impractical violations.

**MAML [4].** A straightforward application of MAML is learning an initialization specialized for EDA. Similar to the protocol of EAML as stated in Section 3.1, we update the whole network in inner loop and outer loop, instead of learning a meta-representation.

**CMA [11].** CMA learns continual manifold for evolving target domain, but does not account for catastrophic forgetting. We use LeNet and conv-4 features as its input.

**DANN+EWC.** EWC [14] is a popular method for continual or incremental learning. We combine EWC regularization with DANN and apply to the EDA problem.

## 4.4 Results

Results on Rotated MNIST are provided in Table 1. Classic domain adaptation methods are not suitable for the evolving target domains. Note that even if we merge the target data as one domain and adapt to it offline (JAN-Merge and CDAN-Merge), the model still cannot capture the structure of evolving target domain. EAML with only meta-representation (EAML-rep) improves performance on all target data and especially the last-adapted target data, since it learns representations specified for evolving target which harness knowledge on the previous target data. The meta-adapter further helps preserve the knowledge learned on early target domains and mitigate forgetting.

We perform ablation study on Evolving Vehicles in Figure 3, validating the models' performance on more complex evolving domains. We test the accuracy on all target data after adapting to a target trajectory in Figure 3(a). EAML-full outperforms all the baselines by a large margin. In Figure 3(b), we show the change of accuracy on the first target domain during the adaptation to the following domains. DANN forgets the knowledge on the previous target data. Note that as EAML-rep adapts to the following target data, the accuracy on the previous target first rises, validating that the meta-representations harness knowledge from previous target when adapting to the current target. EAML-full further incorporates the meta-adapter to overcome catastrophic forgetting.

In Figure 3(c), we depict the accuracy of all target data in meta-testing after adapting to the target trajectories on Caltran. Results indicate that when applied to real-world data, the proposed method still learns effective meta-representations for adapting to evolving target domains.

## 4.5 Analysis

**Meta-representation visualization.** We use t-SNE [37] to visualize the representations learned by different methods on Caltran in Figure 4. The target data evolve as the color changes from dark to light. Training the model only on the source domain results in poor alignment inside the evolving

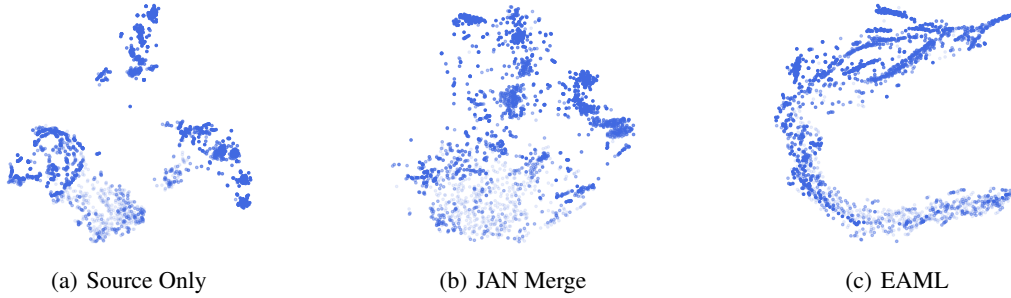

(a) Source Only        (b) JAN Merge        (c) EAML

Figure 4: **T-SNE on Caltran.** The target features evolve as the color changes from dark to light. Training on only the source domain and JAN Merge do not capture the evolvement of target data. EAML learns meta-representation to evolve the target data smoothly and enable continual transfer.

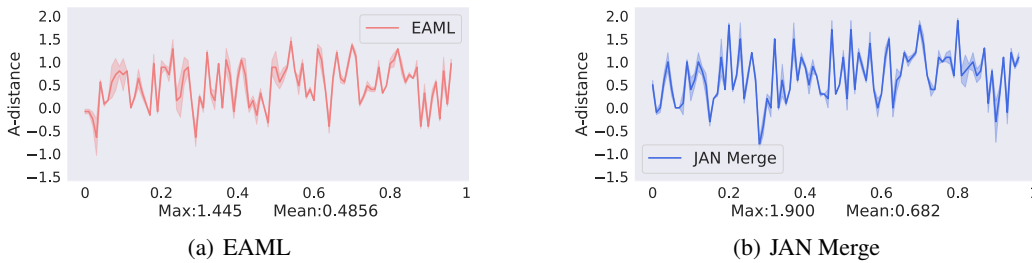

(a) EAML        (b) JAN Merge

Figure 5: $A$-**distance along evolving target domains.** We approximate $\alpha$ in Theorem 1 with $A$-distance between neighboring target data along the evolving target domains on Caltran dataset. EAML achieves smaller maximum and mean values of $A$-distance than JAN Merge, indicating smaller $\alpha$ and smoother evolving representations for the evolving target domain.

target domains as shown in Figure 4(a). Merging the evolving target and adapting to it offline (see Figure 4(b)) bridges the domain shift to some extent, but it does not capture and harness the smooth evolvement of the evolving target domains for knowledge transfer. Figure 4(c) demonstrates the meta-representations of EAML. As expected, EAML captures the evolvement of the target domain and learns a representation that changes smoothly with the evolvement.

$A$-**distance along evolving target domains.** We further assess the representations by considering $\alpha$ in Theorem 1. We approximate $\alpha$ with the proxy $A$-distance [1] between neighboring target data: $d_A(Q_t, Q_{t+0.01}) = 2(1 - 2\epsilon)$, where $\epsilon$ is the generalization error of training a linear classifier to discriminate samples in $Q_t$ and $Q_{t+0.01}$. We plot the changing of $d_A$ with $t \in [0, 0.99]$ in Figure 5. The meta-representation of EAML has much smaller $d_A$ along the evolving target than representations learned by JAN Merge, indicating smaller $\alpha$ and better generalization for EDA.

**Balancing domain adaptation and learning without forgetting.**
The balance of adaptation and avoiding forgetting relies on a hyperparameter $\lambda$ as in line 7 of Algorithm 1 and the learning rate of meta-adapters $\eta_{\phi'}$. Intuitively, larger $\lambda$ and $\eta_{\phi'}$ indicate larger penalty on forgetting. We visualize performance on Caltran w.r.t $\lambda$ and $\eta_{\phi'}$ in Figure 6. Results indicate that EAML is not sensitive to those hyperparameters. The balance between domain adaptation and learning without forgetting is easily achievable.

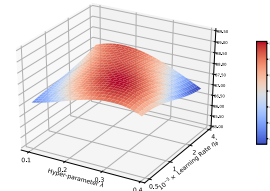

Figure 6: Accuracy on Caltran with various $\lambda$ and $\eta_{\phi'}$.

**Comparison with DANN+Replay.** Generative replay is a powerful method of supervised continual learning. We apply it to evolving domain adaptation on Rotated MNIST as a strong baseline. We train a CycleGAN [41] to transfer images from the source domain to previous target data. Then we add previous target data generated by the CycleGAN when DANN adapts to current target data to mitigate forgetting. Note that this method violates the EDA setting: we wish to deploy light-weight model in meta-testing, yet GANs require heavy computation. We show the performance of DANN+Replay in the last line of Table 1. Although EAML uses only light-weight adapters in meta-testing, it still outperforms DANN+Replay.

Table 2: Comparison of different settings of continual of evolving domain adaptation.

| Setting | Online | Evolvement | Small batch | Forgetting |
|---|:---:|:---:|:---:|:---:|
| CMA [11] and IEDA [2] | ✓ | ✓ | ✓ | × |
| Bobu et al. [3] | × | × | × | ✓ |
| Lao et al. [15] | ✓ | × | × | ✓ |
| **EDA** | ✓ | ✓ | ✓ | ✓ |

## 5    Additional Related Work

**Classic Domain Adaptation.** Classic domain adaptation learns a representation where the domain discrepancy is minimized. [24, 7, 8] map the source and target domains into a new feature space. [19, 36] incorporate the maximum mean discrepancy (MMD). DANN [6] trains a domain discriminator to distinguish discrete source and target while the features are learned adversarially to confuse the discriminator. [35, 20, 34] further improve DANN and achieve significant performance gain. MCD [31] uses an alternative approach: maximizing the disagreement of two classifiers on the target domain. [18, 32, 12] enable pixel-level adaptation with generative architectures. These methods are tailored to discrete source and target domains and cannot be applied to EDA directly.

**Continuous Domain Adaptation.** The problem of continuous domain adaptation is related to EDA but with a different context and focus. [3, 15] learn discrete target tasks online, but they do not address the evolving nature of target domains to harness it in knowledge transfer. [11, 2] focus on continually shifting target domains, but they test the model online and did not address the catastrophic forgetting. In this paper, we tackle a more challenging and realistic EDA scenario, addressing both challenges of target domain evolvement and catastrophic forgetting. A comparison of several different settings of continual or evolving domain adaptation is provided in Table 2.

## 6    Conclusion

In this paper, we address the problem of adapting to evolving domains. We propose a meta-adaptation framework to solve evolving domain adaptation (EDA) efficiently. By learning a meta-representation specified for EDA, we are able to capture and harness the smooth evolvement of the target domain in knowledge transfer. We further overcome the catastrophic forgetting by learning a meta-adapter for weighted feature mimicking. Experiments validate the efficacy of our method on EDA benchmarks.

This paper also opens up future questions for evolving domain adaptation. How can we capture the intrinsic structure of evolving data more efficiently? Could we extent the EDA framework to heterogeneous transfer learning? Furthermore, we hope our work inspires further studies to pursue real-world domain adaptation applications.

## Broader Impact

The propose method may open up the applications of domain adaptation in more real-world scenarios. Without access to target labels, it may help protect privacy. As a method of mitigating the negative effect of dataset shift, EAML can also be applied to undo dataset bias in fair machine learning.

## Acknowledgments and Disclosure of Funding

This work was supported by the Natural Science Foundation of China (61772299, 71690231), Beijing Nova Program (Z201100006820041), and University S&T Innovation Plan by the Ministry of Education of China.

## Footnotes

*Corresponding author: Mingsheng Long (mingsheng@tsinghua.edu.cn)

[2]Codes are available at https://github.com/Liuhong99/EAML.

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
