[Supplementary Material]

# 1 Proof of Theorem 1.

**Theorem 1** (Generalization bounds of evolving domain adaptation). *We further assume $d_{\mathcal{H}\Delta\mathcal{H}}(Q_{t_1}, Q_{t_2}) \leq \alpha|t_1 - t_2|$ holds with constant $\alpha$ for $t_1, t_2 \geq 0$. Then for any $\theta$, with probability at least $1 - \delta$ over the sampling of target trajectory $t_1, t_2 \cdots t_n$,*

$$\mathbb{E}_t \mathbb{E}_{Q_t} L(f_\theta(x), y) \leq \mathbb{E}_P L(f_\theta(x), y) + \frac{1}{n} \sum_{i=1}^{n} [d_{\mathcal{H}\Delta\mathcal{H}}(P, Q_{t_i})] + \mathbb{E}_t \lambda_t + O\left(\frac{\alpha}{\delta n}\right).$$

Recall the mathematical tools in analyzing generalization error bound of classic domain adaptation [1, 5]. Suppose $\mathcal{H} = \{f_\theta | \theta \in \Theta\}$. Then for any distribution $P$ and $Q$, denote by $d_{\mathcal{H}\Delta\mathcal{H}}$ the $\mathcal{H}\Delta\mathcal{H}$-divergence,

$$d_{\mathcal{H}\Delta\mathcal{H}}(P, Q) = \sup_{\theta, \theta' \in \Theta} |\mathbb{E}_P L(f_\theta(x), f_{\theta'}(x)) - \mathbb{E}_Q L(f_\theta(x), f_{\theta'}(x))|,$$

which quantifies the discrepancy between the source and target distributions. The adaptability quantified by $\lambda$ measures the possibility of cross-domain learning [1].

$$\lambda = \min_\theta [\mathbb{E}_P L(f_\theta(x), y) + \mathbb{E}_Q L(f_\theta(x), y)].$$

Then we have the following generalization error bound of classical domain adaptation [1],

**Lemma 1** (Generalization bounds of classic domain adaptation). *For a symmetric loss function $L$ satisfying triangle inequality, and any $f_\theta \in \mathcal{H}$, the following holds,*

$$\mathbb{E}_Q L(f_\theta(x), y) \leq \mathbb{E}_P L(f_\theta(x), y) + d_{\mathcal{H}\Delta\mathcal{H}}(P, Q) + \lambda.$$

Integrating over $t \sim U(0, 1)$, the left hand side becomes the loss on the evolving target domain $Q_t$,

$$\mathbb{E}_t \mathbb{E}_{Q_t} L(f_\theta(x), y) \leq \mathbb{E}_P L(f_\theta(x), y) + \mathbb{E}_t [d_{\mathcal{H}\Delta\mathcal{H}}(P, Q_t)] + \mathbb{E}_t \lambda_t \tag{1}$$

To extend this analysis to the EDA problem, we need to approximate the domain discrepancy $\mathbb{E}_t [d_{\mathcal{H}\Delta\mathcal{H}}(P, Q_t)]$ with finite target trajectories $\{Q_{t_i}\}_{i=1}^n$. Formally, we have the following lemma,

**Lemma 2** (Discretization of $d_{\mathcal{H}\Delta\mathcal{H}}$ on evolving target domain). *Suppose we have target trajectories $\{Q_{t_i}\}_{i=1}^n$ of length $n$. Each $t_i$ is sampled i.i.d. from $U(0, 1)$. Assume $d_{\mathcal{H}\Delta\mathcal{H}}(Q_{t_1}, Q_{t_2}) \leq \alpha|t_1 - t_2|$ holds with constant $\alpha$ for $1 \geq t_1, t_2 \geq 0$. Then with probability at least $1 - \delta$ over the sampling of $t_i$,*

$$\mathbb{E}_t [d_{\mathcal{H}\Delta\mathcal{H}}(P, Q_t)] \leq \frac{1}{n} \sum_{i=1}^{n} [d_{\mathcal{H}\Delta\mathcal{H}}(P, Q_{t_i})] + O\left(\frac{\alpha}{\delta n}\right).$$

*Proof.* Without loss of generality, assume $t_1 \leq t_2 \cdots t_{n-1} \leq t_n$. Denote by $A$ the event $\exists t_i, t_{i+1}, |t_i - t_{i+1}| \geq \frac{2}{m}$. Suppose $D_i$ is the interval $(\frac{i-1}{m}, \frac{i}{m}]$, for $i = 1, 2 \cdots m$. Denote by $B$ the event $\exists i \in \{1, 2 \cdots m\}, \forall j \in \{1, 2 \cdots n\}, t_j \notin D_i$. Then the following holds,

$$P(A) \leq P(B) = 1 - \frac{\binom{n-1}{m-1}}{\binom{n+m}{m-1}}$$

$$\approx 1 - \frac{(n+1)^{n+1}(n-1)^{n-1}}{(n-m)^{n-m}(n+m)^{n+m}}$$

$$\approx 1 - \left(\frac{n-1}{n+1}\right)^{m-1}$$

$$< e^{-\left(\frac{n-1}{n+1}\right)^{m-1}},$$

for large $m$ and $n$. The first step comes from Stirling's equation. The second step is due to $\lim_{x\to 0}(1+x)^{\frac{1}{x}} = e$. The final step holds since $1 + x < e^x$ for $x \neq 0$. Setting $m$ to $\lfloor \frac{1}{2\delta} \rfloor$, we have with probability at least $1 - \delta$,

$$\max_i |t_i - t_{i-1}| < \frac{1}{\frac{\log(1-\delta)}{\log(n-1) - \log(n+1)} + 1} = O\left(\frac{1}{\delta n}\right)$$

Then $\forall t \in (\frac{t_{i-1}+t_i}{2}, \frac{t_i+t_{i+1}}{2}]$,

$$d_{\mathcal{H}\Delta\mathcal{H}}(P, Q_t) \leq d_{\mathcal{H}\Delta\mathcal{H}}(P, Q_{t_i}) + \alpha|t - t_i|$$
$$= d_{\mathcal{H}\Delta\mathcal{H}}(P, Q_{t_i}) + O\left(\frac{\alpha}{\delta n}\right).$$

Integrating over $t$, we complete the proof. $\qquad\square$

To prove Theorem 1, we plug Lemma 2 into equation (1).

## 2  Dataset Details

**Rotated MNIST**: We randomly rotate the MNIST training set and test set by $0 - 180°$. The original training set (60000 images) is used as the source training dataset. For each rotation in the evolving target dataset, we randomly sample only 100 images. In meta-training, the model is trained on the source dataset and random target trajectory with length 10. In meta-testing, for efficient evaluation, the model is adapted sequentially to $120°, 126° \cdots 174°$. Each target rotation in meta-testing also has 100 samples, and we test the performance on the 10000-sample rotated MNIST test set.

**Evolving Vehicles** comprises of 2000 images of sedans and trucks in $1970 - 2020$ collected from `bing.com`. Each decade has 200 sedans and trucks. We use 1980 to 1995 in meta-training, and test on 2000 to 2015 in meta-testing. The batch size is set to 20.

Figure 1: Examples of the Evolving Vehicles dataset.

**Caltran**: The dataset is available at `http://cma.berkeleyvision.org`. See Figure 2 for examples of the dataset. The dataset consists of $5432$ images in total. We use the first $500$ images as the source dataset. We use the following 2000 images in meta-training and the last 2932 images in meta-testing. The batch size is set to 20.

Figure 2: Examples of the Caltran dataset. It contains images captured at an interval of 3 minutes over two weeks, formulating a challenging continually evolving target, since it includes changes in time, illumination, weather, etc.

## 3  Additional Experimental Details

We implement our model on PyTorch with 2080Ti GPUs. We calculate joint MMD following [4]. Suppose the batch size of the source domain is $n_P$ and the batch size of the target domain is $n_Q$,

$$\widehat{d}(P, Q) = \frac{1}{n_P^2}\sum_{i,j} k((x_i^p, \hat{y}_i^p), (x_j^p, \hat{y}_j^p)) + \frac{1}{n_Q^2}\sum_{i,j} k((x_i^q, \hat{y}_i^q), (x_j^q, \hat{y}_j^q)) - \frac{2}{n_P n_Q}\sum_{i,j} k((x_i^p, \hat{y}_i^p), (x_j^q, \hat{y}_j^q)).$$

The feature used in joint MMD is the output of the adapter. The kernel we apply is multi Gaussian kernel based on `https://github.com/thuml/Xlearn/blob/master/pytorch/src/loss.py`.

For Rotated MNIST, the input size of image is $28 \times 28$. We do not adopt further pre-processing to avoid affecting the rotation. For Evolving Vehicles and Caltran, the input size is set to $84 \times 84$. We use random horizontal flip and random resized crop as pre-processing.

The implementation of MAML is based on `https://github.com/dragen1860/MAML-Pytorch/`. In meta-training, we update the adapter and the classifiers for 10 steps in the inner loop. The hyperparameter of weighted adaptation is set with importance weighted cross validation [6].

When implementing the baseline methods, we follow the protocol of DANN [2] and CDAN [3]. The domain discriminator is a three-layer fully connected network with BatchNorm and ReLU activations. The code of CycleGAN is modified based on `https://github.com/junyanz/pytorch-CycleGAN-and-pix2pix`.

**Numerical results in Figure 3 of the main text.** In the main text, we provided results on Caltran and Vehicles in Figure 3 due to limitation of space. We further provide numerical results in Table 1.

Table 1: Accuracy (%) on Evolving Vehicles.

| Method | 1995 | 2000 | 2005 | 2010 | 2015 | 2020 |
|---|---|---|---|---|---|---|
| DANN | 51.0±1.5 | 54.9±0.8 | 58.6±1.2 | 65.5±1.0 | 69.3±1.1 | 70.4±1.5 |
| JAN Merge | 67.2±1.8 | 68.4±1.6 | 66.0±0.9 | 65.7±1.0 | 64.9±1.4 | 64.3±1.0 |
| **EAML** | **70.1**±1.3 | **69.8**±0.9 | **72.5**±0.8 | **73.6**±1.3 | **75.5**±1.4 | **75.2**±1.1 |