[Reviews · NeurIPS 2020]

Review 1

Summary and Contributions: The paper tackles the problem of domain adaptation in which no target labels are observed. The authors proposed an unsupervised meta-adaptation framework EAML that is claimed to deal with domain shift and catastrophic forgetting effectively. Experiments back their claim up.

Strengths: - The paper claims that EAML has applications in continuous domains, which enables EAML to have a broader impact on practice. - The methods are described in detail for each hierarchical step. - Theorem 1 is sound from the perspective that it is later used for representation learning (by using the upper bound).

Weaknesses: - 3 datasets are employed but only for MNIST were the quantified results presented in detail. The ablation study for Vehicles and t-SNE for Caltran are useful, yet cannot be substituted for such results. Likewise, the authors might need to make it clearer as to why such results are missing. Similarly for ablation study and t-SNE, the authors might want to present results for other datasets in the supplemental materials as well. - Figure 5 seems not to clearly show the superiority of EAML vs. JAN Merge. Maybe plotting in the same chart would make things easier to observe.

Correctness: The derivations look correct. For experiments, however, MNIST and Vehicles datasets being used in the paper seem not continuous enough. Maybe if the authors present experiments with more continuous datasets, the correctness of the paper and its claim would be more sound.

Clarity: The paper is written clearly and easy to follow any points they are making.

Relation to Prior Work: The authors discussed the prior work, which covers classic discrete-domain methods and continuos domain adaptation with differences from this work.

Reproducibility: Yes

Additional Feedback: Post Rebuttal: I acknowledge your great effort in providing detailed responses to all reviewers' concerns and would like to increase my score.


Review 2

Summary and Contributions: This paper presents a new domain adaptation setting and proposes an domain adaptation framework based on meta learning to address that. In the new setting the target domain unlabeled data evolves over time and the model is required to adapt to continually evolving target domain without forgetting. Authors proposed the Evolution Adaptive Meta-Learning (EAML) framework, which includes a meta-objective to learn representations for adaptation to continually evolving target domain data, and a meta-adapter for adapting to current target without forgetting the previous target. Experiments demonstrate the effectiveness of the proposed method.

Strengths: + A new practical domain adaptation setting which combines domain adaptation and continual learning. + A novel meta learning framework which can capture the structure of evolving target domain and conduct adaptation to continually evolving target domain data without catastrophic forgetting. + Insightful analysis on the learned meta-representation.

Weaknesses: - It is not clear how to balance domain adaptation and learning without forgetting. - In figure 3, why does JAN Merge underperform EAML-full and EAML-rep at the beginning? Is hyperparameters well tuned? - There is no comparison with other continuous domain adaptation work such as [2,10,14]. - No discussioni with a very relevant work [a]. How does meta-adapter proposed in this paper compared to the AdaIN used in [a]? Z. Wu, et al., ACE: Adapting to Changing Environments for Semantic Segmentation, ICCV 2019. - How to address the case when there is large intra-variance within target domain and there could be abrupt change over time?

Correctness: Yes, sounds correct.

Clarity: It is clear and well organized.

Relation to Prior Work: Most relevant work is discussed but one very relevant work is missing. Z. Wu, et al., ACE: Adapting to Changing Environments for Semantic Segmentation, ICCV 2019.

Reproducibility: Yes

Additional Feedback:


Review 3

Summary and Contributions: The paper aims to study evolving domain adaptation where the domain continuously adapts over time. The paper proposes EAML, Evolution Adaptive Meta Learning which seems to be inspired by MAML by performing representation updates at the outer loop and adapt the adapter module and classifier at the inner loop. In addition, the paper proposes using a meta-adapter (adapter over the adapter modules) to overcome overfitting.

Strengths: - Using more realistic benchmarks (Evolving Vehicles and Caltran) in addition to the simpler rotated MNIST - Moderate novelty. There are multiple levels of meta-learning proposed. - The results seem quite interpretable with the tsne visualization showing the evolution of representation.

Weaknesses: - The model does not consistently perform better than baselines for different settings (e.g. Table 1) - More discussion / comparison with other continual learning literature needed.

Correctness: Seems reasonable.

Clarity: Mostly. I wish the paper describes the baselines in a bit more details. I do not feel that I have a good grasps of how the baselines handle the adaptation.

Relation to Prior Work: I think it is quite clear from how the model is proposed. However the paper can better mention other related work that inspires the proposed approach. For instance, the paper is an adaptation of MAML with modified inner loop that trains only the adapter and the classifier, as well as other meta components. I feel that this is somewhat inspired by Raghu et a. 2019 which shows that adapting the inner loop with only the classifier is quite sufficient. (although the paper adapts the adapter network too, which is slightly different) Overall the related work section is quite abridged. There has been much attention on continual learning recently so I believe it should be expanded more to give additional context of what the literature has done.

Reproducibility: Yes

Additional Feedback: The broader impact section needs much more consideration.


Review 4

Summary and Contributions: This paper studies the problem of domain adaptation with evolving target domain. - A new problem setting, evolving domain adaptation (EDA), is proposed. - A meta-adaptation framework which enables the learner to adapt to continually evolving target domains without forgetting.

Strengths: - The problem of evolving domain adaptation (EDA) is important and practical, yet under explored. - The proposed framework is elegant and logical. - The ablation study in Figure 3 provides many insights into the approach.

Weaknesses: - Problem setting: The motivation behind the combination of domain adaptation and online learning can be further elaborated. Online learning is mainly used when it is computationally infeasible to train over the entire dataset. In domain adaptation, only very few examples are used for the adaptation process. Therefore it is usually feasible to simply store all the data and train in an off-line manner. - Performance: It is unclear how the proposed online setting compares against the offline setting (upperbound), but I suspect that the margin is big. As shown in Table 1, the performance of the proposed framework is weak, making it impractical to choose the online learning setting. - Experiments: Comparison with existing incremental/online learning techniques is missing. It would be interesting to see baseline methods that combine existing domain adaptation and incremental learning techniques. - Comparison with previous work: More comments are provided in the "Relation to prior work" section.

Correctness: Yes. The claims, method, and the expirical methodology seem correct.

Clarity: Yes, the paper is well written overall. Suggestions have been provided in the "additional feedback" section below.

Relation to Prior Work: The difference with previous contributions are not clearly discussed. - Several previous papers on domain adaptation with evolving domains are not cited or compared in the experiments. These papers include "Incremental Adversarial Domain Adaptation for Continually Changing Environments" and "Incremental Evolving Domain Adaptation". - The difference with other continuous domain adaptation methods is unclear. It seems that previous approaches on continuous domain adaptation can be adapted to the proposed evolving domain adaptation (EDA) setting. - The authors did not compare numerically against previous meta-learning-based techniques such as MAML.

Reproducibility: Yes

Additional Feedback: - In line 116-118: the notations f_theta, h_theta are a bit confusing. - The meaning of "rep", "ada", and "full" should be explained in the caption of Table 1. They are not explained until Section 4.4. - Details for network architectures should be provided for reproducibility.

[Author Response · NeurIPS 2020]

Many thanks to all reviewers for your constructive and insightful comments. We address your concerns as follows.

**Response to common questions**

**Q1:** More comparison with other continual/incremental learning literature.

**A1:** We compared with *DANN+Replay* as a continual learning method (in Table 1 of the main text). We add *DANN+EWC* and *DANN+GEM* in Table 3. We will elaborate on continual/incremental learning literature in the revision.

**Q2:** Comparison with other evolving/continual domain methods including [2], [10], [14].

**A2:** Existing continuous or evolving domain refers to **different settings** though in the name of "continuous DA", thus they cannot be adapted to our settings directly. See the *comparison of these settings* in Table 1. Nonetheless, we adapt *CMA [10]* and *Incremental Evolving Domain Adaptation (IEDA)* as evolving domain methods and provide results in Table 3. Due to the missing details and codes of the preprint of [14], we follow their idea and design *DANN+Replay* (Replay Oracle of Table 1 in the main text). We will discuss these settings and methods in detail.

Table 1: Comparison of settings in related work.

| Method | Online | Evolvement | Small-Batch | Forgetting |
|---|---|---|---|---|
| CMA [10] and IEDA | ✓ | ✓ | ✓ | ✗ |
| Bobu *et al.* [2] | ✗ | ✗ | ✗ | ✓ |
| Lao *et al.* [14] | ✓ | ✗ | ✗ | ✓ |
| **EAML** | ✓ | ✓ | ✓ | ✓ |

**Response to Reviewer 6**

**Q1:** Missing quantified results on Caltran and Vehicles.

**A1:** In the main text, we provided results on Caltran and Vehicles in figures due to limitation of space. We further provide numerical results in Table 2 and will add it to the next version.

**Q2:** MNIST and Vehicles are not continuous enough.

**A2:** On MNIST and Vehicles, we sample *random* angles in $0° - 60°$ and years in $1980 - 1995$ during meta-training so that they are *sufficiently continuous*. In meta-testing, we use the same target trajectory on each method for fair and convenient comparison and set the trajectory to $\{120°, 126° \cdots 180°\}$ in MNIST and $\{2000, 2005, 2010, 2015\}$ in Vehicles. To further address the concern, we provide the *randomly sampled meta-testing results* in Table 3.

Table 2: Accuracy (%) on Vehicles.

| Method | 1995 | 2000 | 2005 | 2010 | 2015 | 2020 |
|---|---|---|---|---|---|---|
| DANN | 51.0±1.5 | 54.9±0.8 | 58.6±1.2 | 65.5±1.0 | 69.3±1.1 | 70.4±1.5 |
| JAN Merge | 67.2±1.8 | 68.4±1.6 | 66.0±0.9 | 65.7±1.0 | 64.9±1.4 | 64.3±1.0 |
| **EAML** | **70.1±1.3** | **69.8±0.9** | **72.5±0.8** | **73.6±1.3** | **75.5±1.4** | **75.2±1.1** |

**Response to Reviewer 7**

**Q1:** It is not clear how to balance domain adaptation and learning without forgetting.

**A1:** The balance of adaptation and avoiding forgetting relies on a *hyper-parameter $\lambda$* as in Line 7 of Algorithm 1 in main text and *the learning rate of meta-adapters $\eta_{\phi'}$*. Larger $\lambda$ and $\eta_{\phi'}$ indicate larger penalty on forgetting. We provide *performance on Caltran* w.r.t $\lambda$ and $\eta_{\phi'}$ in Figure 1.

**Q2:** Why does JAN-Merge underperform EAML-full and EAML-rep at the beginning?

**A2:** JAN-Merge merges all target data and adapts to them. Without capturing the evolvement of target data, adapting to them as a whole results in *conflict among evolvement* and hurts the performance on the early-adapted target domains. In Figure 3(b) of main text, EAML performs similarly to DANN at the beginning, but JAN-Merge underperforms EAML and DANN.

**Q4:** Comparison with "ACE: Adapting to Changing Environments for Semantic Segmentation".

**A4:** This reference deals with segmentation, which is a different setting and cannot be compared with our work directly. We will discuss it in detail in a future version.

**Q5:** How to address large intra-variance within target domain and abrupt change?

**A5:** $\alpha$ in Line 83 of main text describes the rate of change in the target. As Theorem 1 points out, the learner generalizes to held-out target data in meta-testing only for reasonably small $\alpha$, which our work falls under. Sudden change in target data is an interesting topic for future work.

Table 3: Rotated MNIST $120° - 180°$ random test.

| Method | Average acc. |
|---|---|
| Source Only | 27.15 |
| DANN | 29.40 |
| CMA [10] | 30.65 |
| IEDA | 30.81 |
| MAML | 31.34 |
| JAN-Merge offline | 31.58 |
| DANN + EWC | 32.66 |
| DANN + GEM | 33.40 |
| **EAML** | **35.03** |

Figure 1: Accuracy on Caltran w.r.t $\lambda$ and $\eta_{\phi'}$.

**Response to Reviewer 8**

**Q1:** The model does not consistently perform better than baselines for different settings.

**A1:** EAML does outperform *all baselines on the average of evolving target data*. For each rotation of MNIST, the performance can be *affected by the sampling of test samples*. Nonetheless, the variation is mostly *within variants of EMAL* and is part of ablation study. We will separate ablation study from quantitative comparison in Table 1.

**Q2:** Details on how baselines perform adaptation.

**A2:** Domain adaptation methods minimize the discrepancy between the source and target features to adapt to the target domain. We modify DANN in EDA by sequentially adapting to target trajectories. JAN-merge and CDAN-merge merge the target trajectories as one target domain and adapt to it *offline*. We will provide more baseline implementation details.

**Response to Reviewer 9**

**Q1:** Motivation behind the combination of domain adaptation (DA) and online learning (OL) can be further elaborated.

**A1:** The proposed setting is *not a simple combination of DA and OL*. Evolving target data are ubiquitous in practice such as changing environment. *Forgetting is intrinsic* in this scenario. Saving target data may lead to *privacy issues*. Besides, evolving target data usually come *in small batches*, making our setting more challenging. We further provide a comparison of different settings of "continual" or "evolving" DA in Table 1.

**Q2:** The performance of the online setting is weak. The comparison against offline upper bound can be big.

**A2:** The performance in Table 1 of the main text seems weak since for each rotation we have *only 100 examples*, making it very challenging. **Offline** methods such as JAN-merge **cannot** be applied to the EDA setting directly, so we test it in offline setting. However, without capturing the information of evolvement, *adapting to target data offline as a whole hurts performance* (also observed in [14]). EAML even outperforms DANN+Replay (Replay Oracle in Table 1 of the main text) which involves heavy training of GANs for privacy-preserving Replay, justifying the efficacy and value of our method.

**Q3:** Comparison with MAML.

**A3:** We compare with *MAML adapted to EDA* in Table 3. Results indicate that the initialization learned with MAML does not provide *enough inductive bias* for EDA.

[Meta-Review · NeurIPS 2020]

After the rebuttal and discussion phase, three reviewers are leaning marginally positive, while reviewer #9 still has concerns. The authors claim that they are studying a new setting which assumes that the target domain is changing rapidly and data from target is arriving in small batches and cannot be stored. R9 is concerned that "(1) the motivation behind proposing the evolving domain adaptation setting, compared to other similar settings (e.g. continuous DA), was unclear, (2) the comparison with baseline methods was not comprehensive - many existing methods can be adapted to the proposed problem with minor modifications." The authors provided a rebuttal where they explain the difference between related work on continuous DA and their setting and provide additional comparisons to existing methods. The rebuttal did not convince R9. After reading the paper and rebuttal, the AC agrees with the authors that there is a difference between their problem setting and other continuous DA work, which is the requirement to avoid catastrophic forgetting on older data as the model adapts to the new data. The AC therefore thinks that the paper can contribute something new to the existing body of research by studying this new problem. The authors should explain this setting difference more clearly in the paper, especially in Figure 1. The authors are also encouraged to include the additional experiments comparing to existing methods for continuous DA in the camera ready.